

# Tolerogenic dendritic cell reporting: Has a minimum information model made a difference?

Ayesha Sahar[1], Ioana Nicorescu[2], Gabrielle Barran[2], Megan Paterson[2], Catharien M.U. Hilkens[2] and Phillip Lord[1]

[1] School of Computing Science, Newcastle University, Newcastle Upon Tyne, United Kingdom
[2] Translational & Clinical Research Institute, Newcastle University, Newcastle Upon Tyne, United Kingdom

## ABSTRACT

Minimum information models are reporting frameworks that describe the essential information that needs to be provided in a publication, so that the work can be repeated or compared to other work. In 2016, Minimum Information about Tolerogenic Antigen-Presenting cells (MITAP) was created to standardize the reporting on tolerogenic antigen-presenting cells, including tolerogenic dendritic cells (tolDCs). tolDCs is a generic term for dendritic cells that have the ability to (re-)establish immune tolerance; they have been developed as a cell therapy for autoimmune diseases or for the prevention of transplant rejection. Because protocols to generate these therapeutic cells vary widely, MITAP was deemed to be a pivotal reporting tool by and for the tolDC community. In this paper, we explored the impact that MITAP has had on the tolDC field. We did this by examining a subset of the available literature on tolDCs. Our analysis shows that MITAP is used in only the minority of relevant papers (14%), but where it is used the amount of metadata available is slightly increased over where it is not. From this, we conclude that MITAP has been a partial success, but that much more needs to be done if standardized reporting is to become common within the discipline.

## INTRODUCTION

A large amount of background information is required to fully understand the context, methods, data and conclusions that pertain to an experiment. Minimum information models (MIMs) provide a framework for reporting all the essential information (metadata) about experimental work and they have become popular among the scientific community. They are designed to make research more reproducible; following their guidelines ensures that the minimum necessary information about the experimental work is presented in a report or publication, so that other researchers can re-use or re-purpose the data and methods.

From 2014 to 2016, a group of researchers working in the field of tolerogenic dendritic cells (tolDCs) came together to generate a MIM for their field. tolDCs are antigen-presenting cells that induce or restore immune tolerance (*Fucikova et al., 2019*). These cells show great

Corresponding authors
Catharien M.U. Hilkens,
catharien.hilkens@newcastle.ac.uk
Phillip Lord,
phillip.lord@newcastle.ac.uk

promise as a therapeutic tool for the treatment of conditions caused by a breach in immune tolerance (*e.g.*, autoimmune diseases) or for the prevention of graft-rejection. However, the diverse range of experimental designs and reagents used to generate tolDCs precluded meaningful comparisons between different tolDC types. It was anticipated that a MIM would improve the transparency, reproducibility and data interpretation of the different tolDC types. MITAP (Minimum Information about Tolerogenic Antigen-Presenting cells) was published in 2016 (*Lord et al., 2016*); the more general name was chosen to allow for the inclusion of other tolerogenic antigen-presenting cell types, for example regulatory macrophages (MRegs).

In total MITAP took 18 months to create as a collaboration by experts in the tolDC field. The whole procedure involved a series of interactive workshops in which several exercises were conducted to gather the relevant basic vocabulary used within the community; to obtain feedback on the draft reporting guidelines, and finally to test its comprehensibility by the end-users. The full MITAP document, including the checklist, can be found on https://w3id.org/ontolink/mitap.

MITAP consists of 4 sections that were considered to be essential by the experts for reporting all necessary information about the generation of tolDCs and other tolerogenic antigen-presenting cell (tolAPC) products. The most crucial stages of the production process are encapsulated in these sections in an orderly manner. They are summarized below:

Section 1. Cells before: This section describes characteristics of the cells before they undergo any manipulation such as (a) essential information about the donor, (b) source of the cells, (c) the cell extraction method (d), cell phenotype after extraction and (e) cell number and viability.

Section 2. Differentiation and induction of tolerogenicity: This section describes the protocol that has been used to differentiate and/or induce tolerogenicity in the cells described in section 1. There are five subsections giving details on (a) preculture conditions, (b) culture conditions, (c) the protocol used to induce differentiation and tolerogenicity of the cells, (d) loading of cells antigen, and (e) the way cells are stored immediately after culture.

Section 3. Cells after: This section describes the characteristics and state of the cells after the differentiation/induction of tolerogenicity process has taken place. This section has three subsections, two of which provide similar basic details as in section 1 on the (a) cell phenotype and (c) cell number and viability. Another part (b) focuses on functional *in vitro* assays such as migratory capacity of the cells, or their ability to induce T regulatory cells.

Section 4. About the protocol: The final section of MITAP describes the general features of the protocol as a whole, such as (a) any external regulatory authorities or guidelines followed; (b) purpose of the cells; (c) whether cell product is applied in an autologous, allogeneic, xenogeneic or syngeneic manner, and finally (d) contact details of the authors.

Here, we examined the usage of MITAP in the tolDC community. More specifically, we addressed the question of whether these reporting guidelines improved the provision of relevant metadata in papers published after the MITAP publication date.

## METHODS AND RESULTS

### How many papers have used MITAP?

The first stage of our analysis was to ask the question of how many papers used MITAP. To answer this, we assumed that any paper that directly used MITAP would include it in the reference list. Therefore, we used Google scholar and downloaded all the papers that had cited MITAP; we found that MITAP had 40 citations in January 2021. By inspection, we found that not all of these papers used MITAP; some papers, for example reviews, referred to MITAP as an example of a MIM. In fact, only 10 out of the 40 papers that had cited MITAP were using it directly as a reporting framework.

### How many papers could have used MITAP?

While it is useful to know the actual penetrance of MITAP, we also wished to understand how many papers that are reporting results about tolDCs could have used MITAP but did not. This question is harder to answer exactly, but we calculated an estimate. Our overall approach is depicted in Fig. 1. We assumed that most of the papers related to tolDCs are available on PubMed. We therefore searched PubMed with a variety of keywords appropriate for papers related to the MITAP subject matter, restricting the search to papers that were published after MITAP itself was published.

Of course, to use the number of search results as an estimate for the total number of papers, we must be sure of our query terms. We achieved this by testing the performance of each query by calculating the precision (how accurate the results were) and recall (how complete the results were). We could test for recall because we had a set of papers that directly cited MITAP; these therefore fall into the category of papers that could (and in this case did) use MITAP. If our query has 100% recall, therefore, all of these papers (and others) should be retrieved. We calculated the precision by simply reading a subset of 20 papers from each query and making a expert judgement about whether they were relevant.

To provide a base line for our queries, we initially generated keywords using inverse document frequency (IDF) from the titles and abstracts of 10 relevant MITAP citations. All the queries were refined by putting a 'date of publication' filter, so that only papers published after MITAP were retrieved, since no papers published before MITAP could have used it. In addition, the review papers were also excluded by applying a ''NOT'' filter on the query, since these were not relevant to this investigation. A query using only keywords that were given in the original MITAP paper retrieved a smaller number of papers, but the recall was also lower (Table 1). Finally, we modified these queries based on expert judgement of the authors resulting in the manual adjusted queries. The precision and recall were higher, but the number of retrieved papers was large. Therefore, the query was further refined by adding keywords such as ''generate'', ''produce'' or ''induce'' to get only those papers that reported on tolDCs that were produced as part of an experiment. The precision and recall of manual adjustment-2 retrieved results were found appropriate among other queries. Thus, this list of 72 papers was decided to be the total number of tolAPC papers published after MITAP (of which the majority focused on tolDCs, as expected). Hence only (10/72) 14% of researchers have used MITAP so far to report their data, while the remaining 62 tolAPC papers did not use MITAP for data reporting.
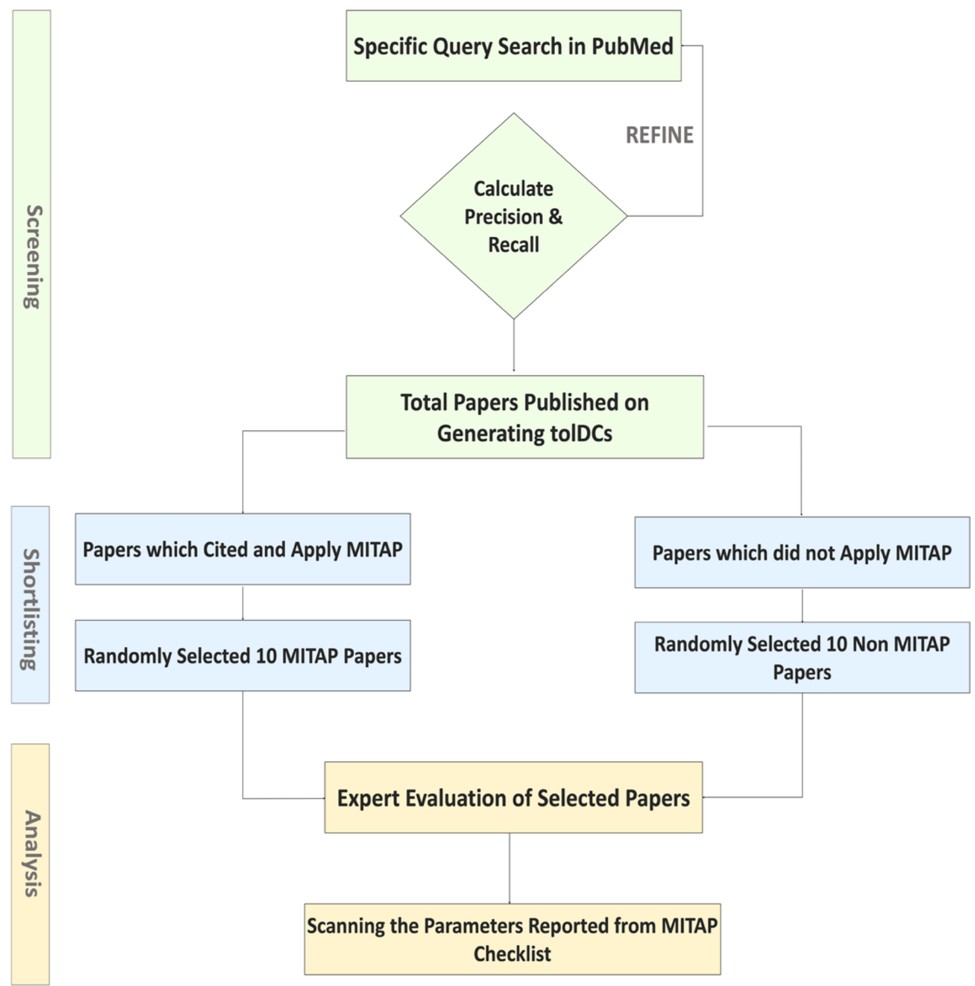

**Figure 1** **The flowchart describing the overall approach of this study.** The above diagram represents the overall approach of this paper. The process begins with specific SQL query based search on PubMed, with further refining the search by using most relevant keywords, a corpus of all the papers published on generating tolDCs is established. An unbiased random selection of 10 papers is performed from each category and these papers are further investigated by experts in the field of tolDCs.

## Comparison of papers with and without MITAP

The purpose of the MITAP document is to ensure that authors provide sufficient basic information about the generation of tolDCs or other types of tolAPC. As shown in the original MITAP paper, complete information is not being provided routinely. 5 years after the MITAP publication, we wanted to re-check the status of information provided by the tolDC community. We therefore applied the MITAP checklist to 10 papers that used MITAP and 10 papers that could have used MITAP but did not. These papers are listed in the Further Reading section, but the sequence in Fig. 2 is different and anonymized. Red sections in the heatmap represent the information missing from the papers, green areas show the information directly provided in the paper and yellow areas show that the information is partially available (*e.g.*, information is available in a referenced paper but
**Table 1  SQL queries on PubMed for retrieving MITAP related publications.** The "*" at the end of words is to represent all versions of that word such as "induce*" refers to induces, induced, induction etc. In addition, a filter to exclude review papers "NOT (review[Title/Abstract])" was also added for all queries.

| | Query | Number of Papers | Precision | Recall |
|---|---|---|---|---|
| **IDF via Titles** | (''tolerogenic dendritic'' **OR** ''derived dendritic'' **OR** ''tolerogenic*'' **OR** ''regulatory cell*'' **OR** ''regulatory-macrophage*'') | 921 | 0.3 | 0.75 |
| **IDF via Abstracts** | (''tolerogenic dendritic'' **OR** ''derived dendritic'' **OR** ''regulatory cell*'' **OR** ''tolerogenic*'' **OR** ''regulatory macrophag*'' **OR** autoimmun* **OR** antigen-presenting) | 8,992 | 0.25 | 0.71 |
| **IDF via MITAP keywords** | ("Tolerogenic dendritic cell" **OR** "Regulatory dendritic cell" **OR** "Tolerogenic antigen-presenting cell" **OR** "Regulatory macrophage" **OR** "Tolerogenic dendritic cells" **OR** "Regulatory dendritic cells" **OR** "Tolerogenic antigen-presenting cells" **OR** "Regulatory macrophages") | 129 | 0.45 | 0.69 |
| **manual adjustment version 1** | ((((''tolerogenic antigen presenting'' **OR** ''tolerogenic-dendritic-cell*'' **OR** ''derived-dendritic*'' **OR** ''regulatory macrophag*'') | 410 | 0.5 | 0.69 |
| **manual adjustment version 2** | (''antigen-presenting'' **OR** ''dendritic cell*'' **OR** ''derived-dendritic*'' **OR** ''tolerogenic*'' **OR** ''regulatory macrophag*'') **AND** (''induc*''[Title] **OR** ''generat*'' **OR** ''develop*''[Title] **OR** ''produc*'') | 72 | 0.75 | 0.83 |

not in the paper itself), or not relevant to the paper. Figure 2 shows that not all the sections of the MITAP checklist were completed by both category of papers. We therefore carried out further analysis to check the percentage of reported parameters for each section and performed statistical analysis. As the data is categorical and does not follow any distribution, the non-parametric McNemar test was carried out, using python 3.9 mlxtend library. For both MITAP and non-MITAP papers, Section 1 is the least reported section with less than 40% of the parameters reported, whereas for all other sections >50% of the parameters were reported. MITAP papers performed marginally better than non-MITAP papers for sections 1, 2 and 4; there were no significant differences for section 3 (Fig. 3 and Table 2).

## Comparison of MITAP with other MIMs

While it is known that high quality data reporting is important and essential for reproducible research, as we have shown, MITAP is currently used by only 14% of tolDC publications. There has been much recent interest on the lack of reproducibility in many areas of biology and immunology, so we doubt that these figures for the tolDC community are unusual. This may be because following standards for the design and subsequent reporting of experiments and subsequent analysis is complex or time consuming. There are a number of ways to reduce the effort of this data reporting; of these, MIMs are seen as an effective mechanism for standardisation, as evidenced by the number of them that being introduced into the medical field (*Fuchs et al., 2018*). Figure 4 shows an increasing trend of MIMs in the medical field in the last 20 years.

Because MITAP was used by a smaller number of published papers than we might have hoped, we analysed and compared usage of other MIMs to MITAP. MITAP was published
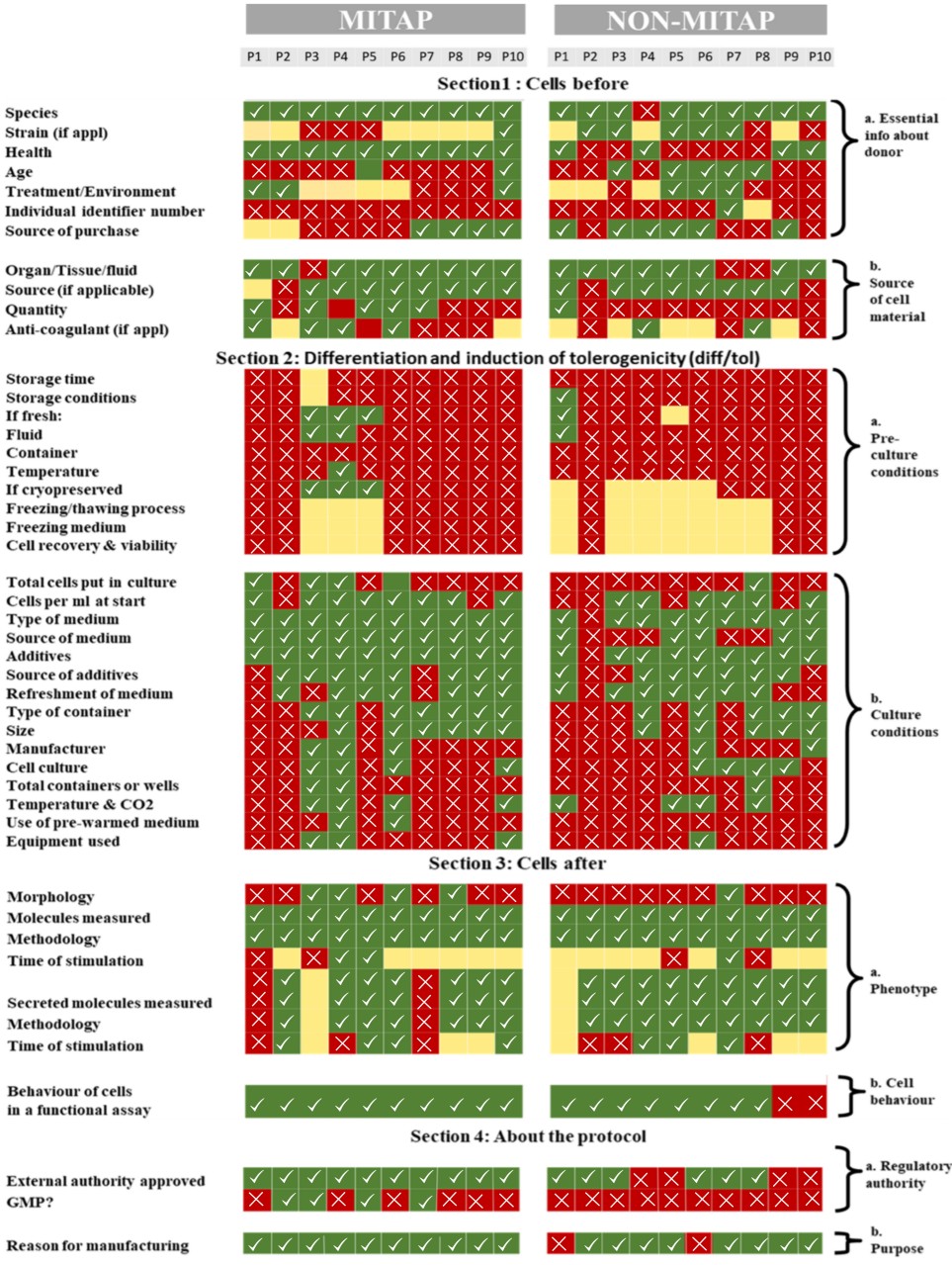

**Figure 2 Heatmap of comparison between MITAP compliant and non-MITAP compliant papers.**
Green/tick boxes: category reported in the publication; yellow/empty boxes: category partially reported in the publication; red/cross boxes: category unreported in the publication. For clear representation, not all subsections are presented in this graph. A complete graph is provided in the Supplementary Data.

in August 2016. Figure 4 shows that five other MIMs were also published in 2016. Looking at the citations of these other five MIMs, MITAP performs to a similar level and is even in the top three most cited MIMs published in 2016 (Fig. 5).
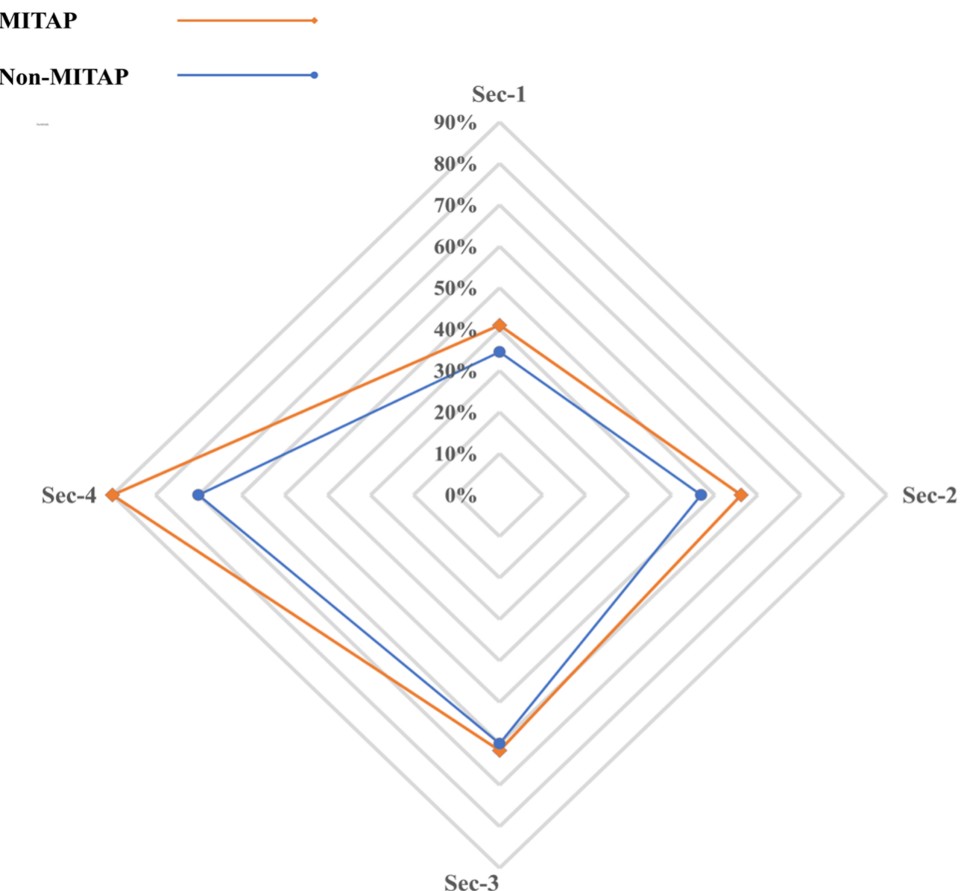

**Figure 3** **Comparison between MITAP and Non-MITAP papers for the total provided categories.** The chart represents comparison between MITAP and Non-MITAP papers for the total provided categories among each section (Sec-1, Sec-2, Sec-3, Sec-4) of the MITAP checklist. Here "total provided" includes the categories fully or partially reported. The further the line is from the center, the higher is the percentage.

In addition to comparing MITAP with MIMs in 2016, it is useful to compare it with other relevant MIMs. MIAME and MIGS are well-established MIMs in the medical field, published in 2001 and 2008, respectively (*Brazma et al., 2001*; *Field et al., 2008*). As dendritic cells are a type of immune cells, we looked at other immunology related MIMs such as MIATA (Minimum Information about T cell Assays) and MITREG (Minimum Information about T Regulatory Cells) (*Britten et al., 2011*; *Fuchs et al., 2018*). Figure 6 shows the citation statistics of these five MIMs compared to MITAP. MIGS and MIAME are MIMs for "omics" data and are both broadly applied in many areas of biology, so might be expected to be widely used, but it can be clearly seen that the citations are still limited. Similarly, MIATA and T cells also cover substantially bigger and older research fields than the tolDC field but show maximum 16 citations only in the peak year.

These citation statistics suggest that MITAP is performing well and is a sufficient tool to implement standardisation in the field of dendritic cells. Even when the MITAP checklist

**Table 2  Significant reporting differences between MITAP and Non-MITAP papers by McNemar's test.**

| Section | Subsection | |
|---|---|---|
| 1 | a | Characteristics of the organism—health |
| | b | Source of cell material—quantity (vol, size, weight) |
| | c | Cell separation—equipment used |
| | | Cell separation—tissue condition between retrieval and cell separation |
| | | Cell separation—methodology |
| 2 | b | Culture conditions—source of medium |
| | c | Differentiation protocol—source of cytokines/other agents |
| | e | Storage—Fluid |
| | | Storage—Container |
| 4 | a | External authority approved |
| | | Does protocol follow GMP? |
| | c | Allogenic/Autologous/Xenogeneic/Syngeneic |

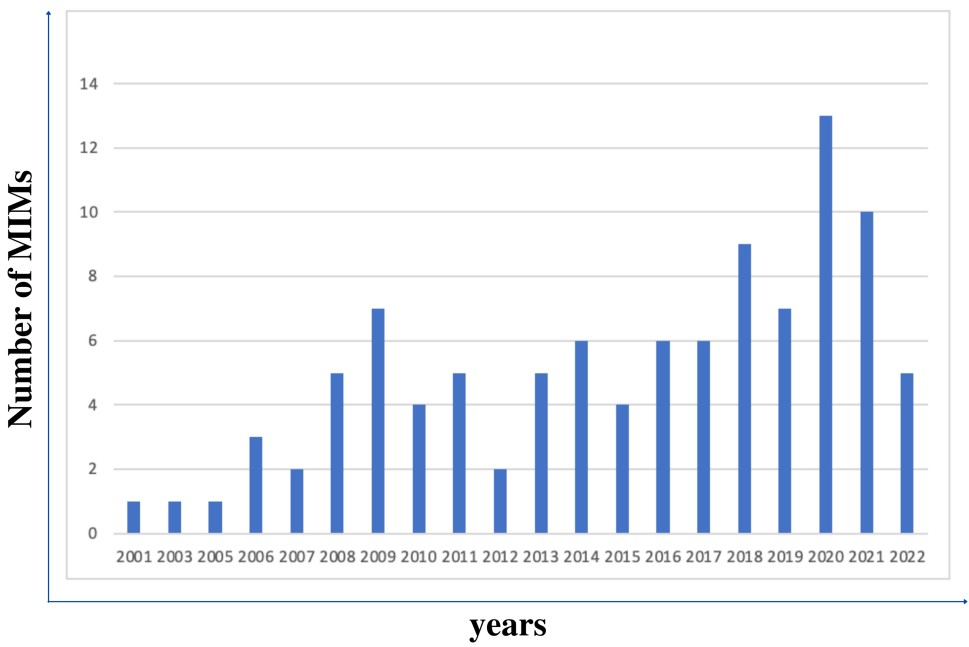

**Figure 4  The figure displays the increasing trend of introducing new MIMs in the biological field over the years.**

is not followed completely, it still fulfils the requirement for providing a minimum set of most important entities for tolDCs, as well as increasing the data reporting.

## DISCUSSION

Over 5 years ago, investigators in the field of tolDC came together to create MITAP, a minimum guidelines tool for reporting of the protocols to generate these cells. By using

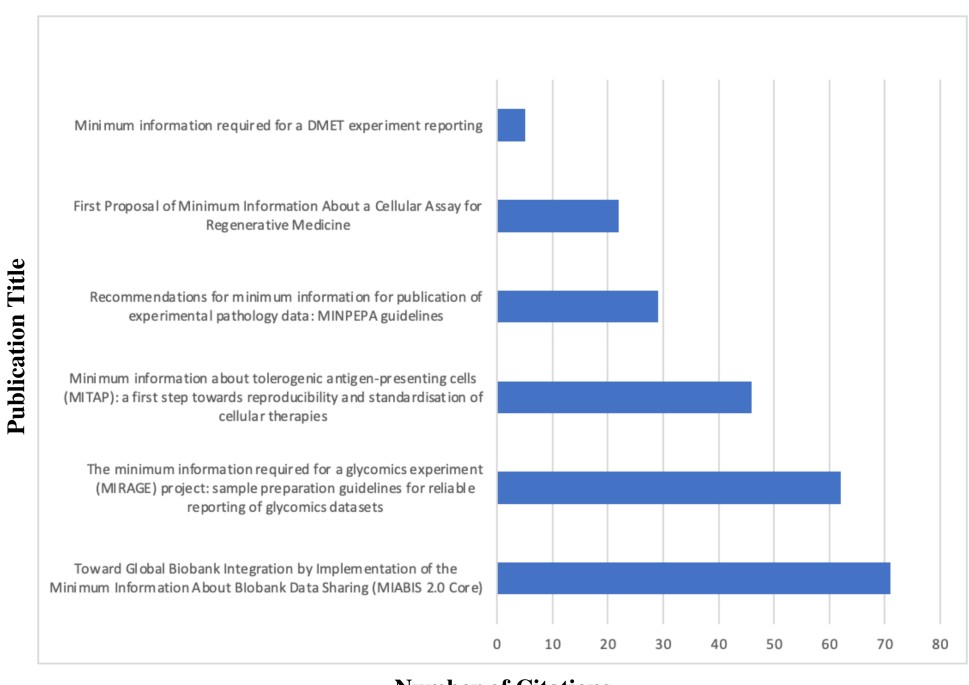

**Figure 5** Comparison of MITAP's performance citation-wise with other related MIM published in the same year (2016). Data Sources: *Kumuthini et al. (2016)*, *Sakurai et al. (2016)*, *Scudamore et al. (2016)*, *Struwe et al. (2016)* and *Merino-Martinez et al. (2016)*.

MITAP on extant papers that were published before MITAP came into existence, we showed that a large proportion of papers lacked sufficient data required to interpret and reproduce the generation of these tolerogenic cells (*Lord et al., 2016*). Here, we investigated usage of MITAP, and whether it improved the description of the four protocol sections: (1) Cells before; (2) differentiation and induction of tolerogenicity; (3) cells after, and (4) about the protocol.

Despite the highly collaborative nature of building MITAP, which involved many experts in the field, we have found that the use of MITAP is surprisingly low: only 14% of research papers that could have used MITAP, actually did. Although encouragingly, those papers that used MITAP performed slightly better on reporting relevant data across three of the four protocol sections, experimental details remained under-reported especially in section 1 ('Cells before') and 2 ('Induction').

There were no significant differences in reporting on section 3 ('Cells after') between MITAP- and non-MITAP papers. This can most likely be explained by the fact that section 3 deals with the final tolDC product itself, as these cells are the main focus of these papers, details on these cells are usually well reported. In contrast, section 1 and 2, which describe the source and features of the cells before (section 1) they undergo the differentiation process to become tolDCs (section 2), were significantly under-reported in non-MITAP papers compared to MITAP papers.
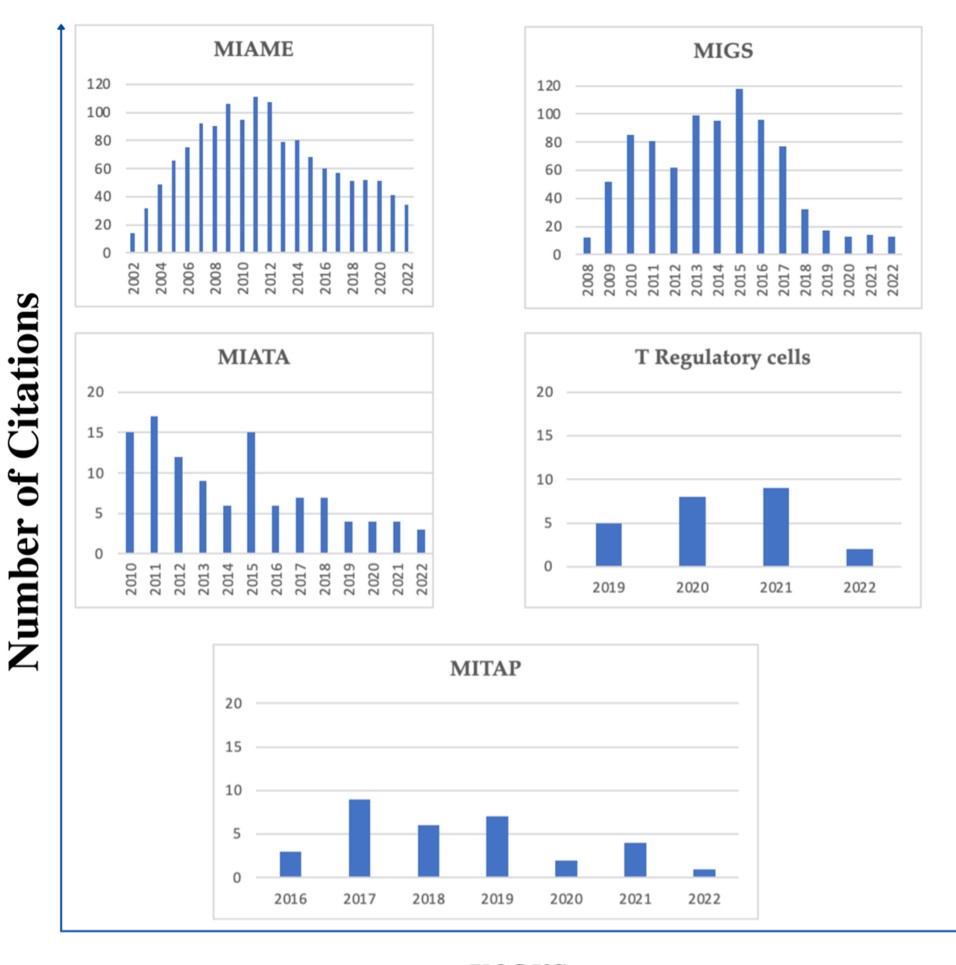

**Number of Citations**

**years**

**Figure 6** **Comparison of MITAP's performance citation-wise with other related MIMs.**

It is not known whether this is because the authors either did not have the relevant information or data available or just did not report it. Nevertheless, under-reporting of these sections will ultimately make it more difficult for investigators to reproduce published data and/or to make comparisons between different types of tolDCs. As it is becoming increasingly clear that considerable heterogeneity exists between different tolDC products, but also between tolDC products derived from different patients' groups (*Navarro-Barriuso, Mansilla & Martínez-Cáceres, 2018*; *Navarro-Barriuso et al., 2019*), it is becoming even more important to improve reporting on the cell source and culture protocol to generate tolDCs.

There is a growing recognition that open data reporting and standardisation are necessary for repurposing or reusing data to discover valuable insights from past work, thus promoting research transparency. In this context, the 14% usage of MITAP might seem low or disappointing. However, our experience shows that the take up of MIMs takes time, despite the general appreciation of their importance. A possible reason for

this could be the disparity between effort and reward: for the person producing the data, standardisation may be time-consuming with low reward; conversely, for those consuming the data, they provide high reward, maximize repurposing and reuse, with much less effort that otherwise; a MIM can provide standardisation and tooling can help to reduce the effort for data producers, but it is a high effort activity for those experts developing the MIMs. We also note that, from an informal analysis of the papers, that MITAP usage seems higher among papers coming from European authors; this perhaps suggests that more outreach work is needed, with scientists in other parts of the world.

Perhaps the main way to increase the penetrance of a MIM is the availability of a relevant data repository. MIAME is one of the most successful MIMs and has relevant repositories such as ArrayExpress or GEO requiring MIAME compliant data submission. As a result, many journals indirectly also require researchers to follow MIAME to submit the data. Other factors to the success of MIM can be the availability of software to record the metadata combined with a database to retrieve the data along with the metadata efficiently.

Our experiments demonstrate that MITAP is sufficient for the reporting of data; as a result, it has fulfilled its key objective. However, to make its use more widespread in the field, we need to introduce an online platform where researchers can record, edit, and save the metadata in a user-friendly manner.

### Funding
This project has received funding from the European Union's Horizon 2020 research and innovation programme under grant agreement No 860003 and BBSRC/GSK Case studentship award BB/S507039/1. The funders had no role in study design, data collection and analysis, decision to publish, or preparation of the manuscript.

### Grant Disclosures
The following grant information was disclosed by the authors:
European Union's Horizon 2020 research and innovation programme: 860003.
BBSRC/GSK Case studentship: BB/S507039/1.

### Competing Interests
The authors declare there are no competing interests.

### Author Contributions
- Ayesha Sahar conceived and designed the experiments, performed the experiments, analyzed the data, prepared figures and/or tables, authored or reviewed drafts of the article, and approved the final draft.
- Ioana Nicorescu performed the experiments, authored or reviewed drafts of the article, and approved the final draft.
- Gabrielle Barran performed the experiments, authored or reviewed drafts of the article, and approved the final draft.

- Megan Paterson performed the experiments, authored or reviewed drafts of the article, and approved the final draft.
- Catharien M.U. Hilkens conceived and designed the experiments, authored or reviewed drafts of the article, and approved the final draft.
- Phillip Lord conceived and designed the experiments, authored or reviewed drafts of the article, and approved the final draft.

## Data Availability

Our study does not produce or use any raw data

## Supplemental Information

Supplemental information for this article can be found online at http://dx.doi.org/10.7717/peerj.15352#supplemental-information.

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

## FURTHER READING

**Anderson AE, Swan DJ, Wong OY, Buck M, Eltherington O, Harry RA, Patterson AM, Pratt AG, Reynolds G, Doran JP, Kirby JA. 2017.** Tolerogenic dendritic cells generated with dexamethasone and vitamin D3 regulate rheumatoid arthritis CD4+ T cells partly via transforming growth factor- $\beta$ 1. *Clinical & Experimental Immunology* **187(1)**:113–23 DOI 10.1111/cei.12870.

**Bouchet-Delbos L, Even A, Varey E, Saïagh S, Bercegeay S, Braudeau C, Dréno B, Blancho G, Josien R, Cuturi MC, Moreau A. 2021.** Preclinical assessment of autologous tolerogenic dendritic cells from end-stage renal disease patients. *Transplantation* **105(4)**:832–841 DOI 10.1097/tp.0000000000003315.

**Dawicki W, Huang H, Ma Y, Town J, Zhang X, Rudulier CD, Gordon JR. 2021.** CD40 signaling augments IL-10 expression and the tolerogenicity of IL-10-induced regulatory dendritic cells. *PLOS ONE* **16(4)**:e0248290 DOI 10.1371/journal.pone.0248290.

**Eslami-kaliji F, Sarafbidabad M, Kiani-Esfahani A, Mirahmadi-Zare SZ, Dormiani K. 2021.** 10-hydroxy-2-decenoic acid a bio-immunomodulator in tissue engineering; generates tolerogenic dendritic cells by blocking the toll-like receptor 4. *Journal of Biomedical Materials Research Part A* **109(9)**:1575–1587 DOI 10.1002/jbm.a.37152.

**Funda DP, Goliáš J, Hudcovic T, Kozáková H, Špíšek R, Palová-Jelínková L. 2018.** Antigen loading (eg. glutamic acid decarboxylase 65) of tolerogenic DCs (tolDCs) reduces their capacity to prevent diabetes in the non-obese diabetes (NOD)-severe combined immunodeficiency model of adoptive cotransfer of diabetes as well as in NOD mice. *Frontiers in Immunology* **9**:290 DOI 10.3389/fimmu.2018.00290.

**Garcia AM, Bishop EL, Li D, Jeffery LE, Garten A, Thakker A, Certo M, Mauro C, Tennant DA, Dimeloe S, Evelo CT. 2021.** Tolerogenic effects of 1, 25-dihydroxyvitamin D on dendritic cells involve induction of fatty acid synthesis. *The Journal of Steroid Biochemistry and Molecular Biology* **211**:105891 DOI 10.1016/j.jsbmb.2021.105891.

**García-González PA, Schinnerling K, Sepulveda-Gutierrez A, Maggi J, Hoyos L, Morales RA, Mehdi AM, Nel HJ, Soto L, Pesce B, Molina MC. 2016.** Treatment with dexamethasone and monophosphoryl lipid A removes disease-associated transcriptional signatures in monocyte-derived dendritic cells from rheumatoid arthritis patients and confers tolerogenic features. *Frontiers in Immunology* **7**:458 DOI 10.3389/fimmu.2016.00458.

**Hutchinson JA, Ahrens N, Geissler EK. 2017.** MITAP-compliant characterization of human regulatory macrophages. *Transplant International* **30(8)**:765–775 DOI 10.1111/tri.12988.

**Ilic N, Gruden-Movsesijan A, Cvetkovic J, Tomic S, Vucevic DB, Aranzamendi C, Colic M, Pinelli E, Sofronic-Milosavljevic L. 2018.** Trichinella spiralis excretory–secretory products induce tolerogenic properties in human dendritic cells via Toll-like receptors 2 and 4. *Frontiers in Immunology* **9**:11 DOI 10.3389/fimmu.2018.00011.

Lee HY, Kim J, Ryu JS, Park SJ. 2017. Trichomonas vaginalis $\alpha$-actinin 2 modulates host immune responses by inducing tolerogenic dendritic cells via IL-10 production from regulatory T cells. *The Korean Journal of Parasitology* **55(4)**:375 DOI 10.3347/kjp.2017.55.4.375.

Lee JA, Spidlen J, Boyce K, Cai J, Crosbie N, Dalphin M, Furlong J, Gasparetto M, Goldberg M, Goralczyk EM, Hyun B. 2008. MIFlowCyt: the minimum information about a flow cytometry experiment. *Cytometry Part A: the Journal of the International Society for Analytical Cytology* **73(10)**:926–930 DOI 10.1002/cyto.a.20623.

Li M, Eckl J, Abicht JM, Mayr T, Reichart B, Schendel DJ, Pohla H. 2018. Induction of porcine-specific regulatory T cells with high specificity and expression of IL-10 and TGF-$\beta$1 using baboon-derived tolerogenic dendritic cells. *Xenotransplantation* **25(1)**:e12355 DOI 10.1111/xen.12355.

Navarro-Barriuso J, Mansilla MJ, Quirant-Sánchez B, Teniente-Serra A, Ramo-Tello C, Martínez-Cáceres EM. 2021. Vitamin D3-induced tolerogenic dendritic cells modulate the transcriptomic profile of T CD4+ cells towards a functional hyporesponsiveness. *Frontiers in Immunology* **2021**:3461 DOI 10.3389/fimmu.2020.599623.

Novère NL, Finney A, Hucka M, Bhalla US, Campagne F, Collado-Vides J, Crampin EJ, Halstead M, Klipp E, Mendes P, Nielsen P. 2005. Minimum information requested in the annotation of biochemical models (MIRIAM). *Nature Biotechnology* **23(12)**:1509–1515 DOI 10.1038/nbt1156.

Perdijk O, Van Neerven RJ, Meijer B, Savelkoul HF, Brugman S. 2018. Induction of human tolerogenic dendritic cells by 3′-sialyllactose via TLR4 is explained by LPS contamination. *Glycobiology*. **28(3)**:126–130 DOI 10.1093/glycob/cwx106.

Phillips BE, Garciafigueroa Y, Engman C, Trucco M, Giannoukakis N. 2019. Tolerogenic dendritic cells and T-regulatory cells at the clinical trials crossroad for the treatment of autoimmune disease; emphasis on type 1 diabetes therapy. *Frontiers in Immunology* **10**:148 DOI 10.3389/fimmu.2019.00148.

Sawitzki B, Harden PN, Reinke P, Moreau A, Hutchinson JA, Game DS, Tang Q, Guinan EC, Battaglia M, Burlingham WJ, Roberts IS. 2020. Regulatory cell therapy in kidney transplantation (The ONE Study): a harmonised design and analysis of seven non-randomised, single-arm, phase 1/2A trials. *The Lancet* **395(10237)**:1627–1639 DOI 10.1016/S0140-6736(20)30167-7.

Song HY, Kim WS, Han JM, Park WY, Lim ST, Byun EB. 2021. HMOC, a chrysin derivative, induces tolerogenic properties in lipopolysaccharide-stimulated dendritic cells. *International Immunopharmacology* **95**:107523 DOI 10.1016/j.intimp.2021.107523.

Spiering R, Jansen MA, Wood MJ, Fath AA, Eltherington O, Anderson AE, Pratt AG, Van Eden W, Isaacs JD, Broere F, Hilkens CM. 2019. Targeting of tolerogenic dendritic cells to heat-shock proteins in inflammatory arthritis. *Journal of Translational Medicine* **17(1)**:1–2 DOI 10.1186/s12967-019-2128-4.

Struwe WB, Agravat S, Aoki-Kinoshita KF, Campbell MP, Costello CE, Dell A, Feizi T, Haslam SM, Karlsson NG, Khoo K-H, Kolarich D, Liu Y, McBride R, Novotny MV, Packer NH, Paulson JC, Rapp E, Ranzinger R, Rudd PM, Smith DF,

**Tiemeyer M, Wells L, York WS, Zaia J, Kettner C. 2016.** The minimum information required for a glycomics experiment (MIRAGE) project: sample preparation guidelines for reliable reporting of glycomics datasets. *Glycobiology* **26(9)**:907–910 DOI 10.1093/glycob/cww082.

**Švajger U, Rožman PJ. 2019.** Synergistic effects of interferon-$\gamma$ and vitamin D3 signaling in induction of ILT-3highPDL-1high tolerogenic dendritic cells. *Frontiers in Immunology* **10**:2627 DOI 10.3389/fimmu.2019.02627.

**Tomić S, Joksimović B, Bekić M, Vasiljević M, Milanović M, Čolić M, Vučević D. 2019.** Prostaglanin-E2 potentiates the suppressive functions of human mononuclear myeloid-derived suppressor cells and increases their capacity to expand IL-10-producing regulatory T cell subsets. *Frontiers in Immunology* **10**:475 DOI 10.3389/fimmu.2019.00475.

**Zhang M, Zheng Y, Sun Y, Li S, Chen L, Jin X, Hou X, Liu X, Chen Q, Li J, Liu M. 2019.** Knockdown of NEAT1 induces tolerogenic phenotype in dendritic cells by inhibiting activation of NLRP3 inflammasome. *Theranostics* **9(12)**:3425 DOI 10.7150/thno.33178.

**Zhou Y, Leng X, Li H, Yang S, Yang T, Li L, Xiong Y, Zou Q, Liu Y, Wang Y. 2017.** Tolerogenic dendritic cells induced by BD750 ameliorate proinflammatory T cell responses and experimental autoimmune encephalitis in mice. *Molecular Medicine* **23(1)**:204–214 DOI 10.2119/molmed.2016.00110.

**Zubizarreta I, Flórez-Grau G, Vila G, Cabezón R, España C, Andorra M, Saiz A, Llufriu S, Sepulveda M, Sola-Valls N, Martinez-Lapiscina EH. 2019.** Immune tolerance in multiple sclerosis and neuromyelitis optica with peptide-loaded tolerogenic dendritic cells in a phase 1b trial. *Proceedings of the National Academy of Sciences of the United States of America* **116(17)**:8463–8470 DOI 10.1073/pnas.1820039116.