# Peer review of "Tolerogenic dendritic cell reporting: Has a minimum information model made a difference?"

_PeerJ, doi:10.7717/peerj.15352_

## Round 0.1 · original submission · Minor Revisions

Both reviewers give opinions on minor revisions, and hope the author can revise them as soon as possible and answer questions from experts.

·

Basic reporting

All four criteria have been met

Experimental design

All four criteria have been met.
A novel and interesting design for audit of metadata.

Validity of the findings

The findings are accurately presented The interpretation might be argued.

Additional comments

General comments

This is an important paper not for the specific of its content ie the use of a minimally defined protocol for a cell therapy (in this case tolerogenic antigen presenting cells) but as a follow-up audit of a recommended best practice model for the conduct of a clinical trial. More specifically the paper audits the use of MITAP (Minimum Information about Tolerogenic Antigen-Presenting cells). Tolerogenic antigen presenting cells are potentially a useful therapy for many immune mediated disease but their general application is obstructed by the diverse and sometimes contradictory methods of application. Not often are such recommendations as MITAP made.
On this occasion, an excellent MITAP for the use of tolerogenic antigen presenting cells has been developed as a consensus document for treatment of various immunological disorders and was presented to the community in 2016. The purpose of the MITAP is to ensure some level of standardisation in clinical trials, especially of tolerogenic dendritic cells, so that data from different trials can be adequately compared. This is clearly important if new therapies are to be safely introduced. This paper reviews the number of studies since that time which have taken consideration of MITAP and followed the recommendations, and compared them with papers which did not follow MITAP procedures.
While the use of MITAP in research papers is a valuable step forward, the uptake rate of 14% by eligible studies is considered by the authors as a good result; However, it still seems to this reviewer to be below expectation.

Specific comments
Why did only 25% of trial follow MITAP? Are the original recommendations to difficult to adhere to?
Appendices with reference to the raw data of papers which used or did not use MITAP are included. Would there be any value in surveying the authors of these papers to invite opinion of why they did or did not follow MITAP?
The methodology is novel and useful eg the heat maps in Fig.2.
The suggestion to set up an online platform for data deposition is excellent.
A difficulty with human tolDC methodology is that, outside of MITAP, there is not yet consensus on optimal source and preparation of "cells before" and `'cells after". The authors report that there was significant under-reporting of data in sections 1 &2 in MITAP papers vs non-MITAP papers, but more important it appears that underreporting in MITAP papers was still not optimal and adherence to MITAP appeared to be low.

Reviewer 2 ·

Basic reporting

This article satisfies basic reporting criteria.

Experimental design

The experimental design is sound.

Validity of the findings

The findings are valid.

Additional comments

Sahar and colleagues report upon the adoption and correct implementation of the MITAP guidelines in articles published since 2016 that describe generation of TolAPCs. Their data collection strategy is well-described and convincing. Using the MITAP checklist to compare the methodological contents of TolAPC-related original research articles that either explicitly followed MITAP with those that did not, led the authors to conclude that (1) penetrance of MITAP was low, and (2) using MITAP only marginally improved reporting on average. This is a valuable report that highlights the on-going challenge of standardizing research in the field. Please would the authors consider the following minor points:

1. The level of non-compliance in articles that claimed to follow MITAP is concerning. To what extent does this reflect, (1) the inability of authors to follow simple guidelines, (2) the over-complexity of MITAP guidelines, or (3) a problem with using an unweighted count of MITAP criteria as a compliance metric?

2. L122 – please state the total number of MITAP non-compliant articles that were returned by your filtered search strategy.

3. Judging from the authors’ names, the MITAP-compliant articles seem to come primarily from European research groups. In contrast, 5 non-compliant articles come from groups with Asian or Middle Eastern author names. Were there any MITAP-compliant articles from North America? Please discuss this possible geographical bias. Could the low penetrance of MITAP be attributed to cultural or language barriers, as opposed to scientific obstacles?

This is a scientifically sound article that addresses an important facet of TolAPC research. I recommend its publication after these minor corrections.

---

## Round 0.2 · accepted · Accept

After modification, both reviewers give suggestions for acceptance. In my opinion, there is basically no publishing risk in this study, which meets the publishing requirements.

·

Basic reporting

see previous review.
The authors have replied satisfactorily to this referee's queries.

Experimental design

see previous review

Validity of the findings

see previous review

Additional comments

An excellent paper, enhanced with additional material included after revision. I have not further comments.

Reviewer 2 ·

Basic reporting

This article meets all criteria for publication.

Experimental design

This article meets all criteria for publication.

Validity of the findings

This article meets all criteria for publication.

Additional comments

This article meets all criteria for publication.